# Use Optimization of Organic Wastes in Anaerobic Soil Disinfestation against Strawberry Charcoal Rot Root

Ana Márquez-Caro, Celia Borrero *, Paloma Hernández-Muñiz and Manuel Avilés

Agronomy Department E.T.S.I.A., Universidad de Sevilla, Utrera road Km 1, 41013 Sevilla, Spain
* Correspondence: cborrero@us.es

**Abstract:** The imminent removal of agrochemicals in Europe is directly affecting the strawberry sector in Spain, specifically in the Huelva province. Huelva is currently the main producer in the country. The lack of efficient fumigants has led to the rise of soilborne diseases. *Macrophomina phaseolina* (the causal agent of charcoal rot in strawberries) is generating serious problems with strawberry farmers. This work proposes to optimize the process of anaerobic soil disinfestation (ASD) against *M. phaseolina* in strawberry crops as a more sustainable alternative than chemical fumigation. Two trials with different inoculum doses were conducted, amending the soil with rice bran and residual strawberry extrudate as the carbon source for the ASD to evaluate the propagule reduction. After the ASD, these soils were used to crop strawberries in a growth chamber under controlled conditions to assay the disease reduction. Strawberry extrudate and rice bran favored disease reduction over the control, equaling the efficacy of Metam Sodium or even improving it in some cases. *Trichoderma* spp. and microbial activity could influence the suppressive effect of the ASD. All amended ASD treatments reduced the *M. phaseolina* propagules and charcoal rot severity. Rice bran and residual strawberry extrudate are suitable amendments to be used in ASD against the pathogen. The resulting soil disinfestation against *M. phaseolina* with residual strawberry extrudate at 16.89 t/ha and 25 days was similar to the most effective treatment with rice bran (20 t/ha and 40 days) based on the disease foliar severity. Additionally, both treatments were equivalent in efficacy to the Metam Sodium treatment.

**Keywords:** β-glucosidase activity; *Macrophomina phaseolina*; residual strawberry extrudate; rice bran; propagule survival; disease severity





## 1. Introduction

Preplant soil fumigation technology is the main method of managing strawberry (*Fragaria × ananassa* Duch.) soil-borne diseases worldwide [1]. Although highly effective, soil fumigation has increasingly been subjected to regulatory restrictions or outright bans due to potential environmental and human health implications [2–4]. In fact, in the European Union, dazomet, Metam Potassium, and Metam Sodium (methyl isocyanate generators) have failed the review of the uniform principles established by Regulation (EC) No. 1107/2009, and the authorization for their use has been restricted. Due to this, there is an urgent need to search for nonchemical disinfestation methods that prevent crop losses but do not cause environmental or health problems [5].

Strawberries are a high-value crop, but their production is limited by numerous pest issues, including a diversity of lethal soil-borne diseases such as charcoal rot, Verticillium wilt, and Fusarium wilt [5]. The adoption of alternative fumigants, due to the elimination of methyl bromide, corresponded with increases in crop losses from emerging diseases caused by *Fusarium oxysporum* f. sp. *fragariae* [6] (Winks and William) and *Macrophomina phaseolina* [7,8] (Tassi) Goid worldwide.

Charcoal rot symptoms include wilting of the foliage, the drying and defoliation of older leaves, and plant death [7]. The appearance of *M. phaseolina* is a problem for strawberry crops in Spain, as this country has 7260 ha of strawberry crops, most of which

are located in Huelva province (South Spain) [9,10]. Huelva owns 25–33% of the total European production and is the world's largest exporter [11]. Charcoal rot caused by *M. phaseolina* has been considered a prevalent disease in Huelva since 2008. Thus, the control of charcoal rot caused by *M. phaseolina* has not yet been achieved in strawberry crops and is a challenge for agricultural research [12,13].

Anaerobic soil disinfestation (ASD) has emerged as an alternative to chemical fungicides. This technique consists of incorporating labile carbon sources into the soil, watering to field capacity, and covering with an impermeable sheet. This situation normally continues for several weeks to achieve reducing conditions [14]. This technique has been examined as a potential alternative to fumigation for soil-borne disease control in numerous cropping systems [15], including strawberries [16,17]. This technique produces constant dynamic changes in the soil microbiology composition associated with disease control [18,19]. In general, the effect of the modified microbiome in disease control is believed to be due to the change-driven transformation of the soil metabolism, so the resulting microbial community interferes with the pathogenic activities [19]. Disease control achieved in response to ASD is mainly due to the production of various active chemicals, such as volatile fatty acids, methyl sulfide compounds, and hydrocarbons with antimicrobial activity [19–21]. The effectiveness of disease control in response to ASD is influenced by many factors, including the soil temperature, organic input ratio C:N, amendment amount, soil moisture content, and the specific pathogen targeted [22].

The main objective of this study was to determine if ASD causes a reduction of *M. phaseolina* propagules. For this purpose, rice bran and residual strawberry extrudate were used as amendments. We also wanted to check if this reduction in the inoculum and the resulting change in the soil microbiome lead to a noticeable reduction in the strawberry disease severity. Different doses of amendment, fungus inoculum, and ASD durations were tested, with the intention of optimizing the ASD treatments.

## 2. Materials and Methods

### 2.1. Soil and Amendments Characterization

The soil was collected in Huelva. This soil showed the typical soil characteristics for growing strawberries in this area. The soil had no precedent for soilborne pathogens, but even so, a microbiological characterization was performed to ensure the absence of these pathogens and check for the presence of *Trichoderma* spp. To perform soil characterization, soil samples were taken from 20 cm deep after the superficial layer was removed. Texture [23], bulk density [14], field capacity, pH [24], Olsen phosphorus [25], electric conductivity (conductivity in extract 1/5), exchange cations [26], available trace elements [27], and concentration of available nutrients [28] were measured in the soil. Carbon and nitrogen contents were also determined in the soil using a LECO TruSpec CN Elemental Analyzer (St. Joseph, MI, USA) (Table 1).

Industrial wastes were chosen as amendments due to their location. The selected amendments were rice bran and residual strawberry extrudate.

Rice bran (ARROZÚA, Sevilla, Spain) was chosen as the standard amendment, as it is the most used in studies of ASD. Bran provides an adequate supply of organic matter to obtain the desired disinfestation effect [29–31].

Residual strawberry extrudate (SVZ Almonte S.A., Huelva, Spain) corresponds to the waste obtained from the production of strawberries that are not marketable due to their condition or quality. The residual extrusions retained in the 2.2-mm and 0.6-mm sieves were selected, and then, a homogeneous mixture of both residues was made to obtain the one that was finally used in the trials. This residue was easily accessible in the Huelva strawberry growing area and contained a significant amount of oxidable carbon, ideal for ASD. Furthermore, the residual strawberry extrudate could reach around 7% by weight of the manufactured strawberries and must be properly managed due to its high organic load [32]. Both residues were kept in cold storage at −20 °C until their use.

**Table 1.** Physicochemical soil properties.

| Parameter | Value † |
|---|---|
| pH (extractor 1/2.5 p/V) | 5.52 ± 0.055 |
| Electrical conductivity (μS/cm) (extractor 1/5 p/V) | 216.20 ± 13.273 |
| Olsen P (mg/kg) | 62.18 ± 1.746 |
| Oxidable organic matter (%) | 0.42 ± 0.005 |
| Oxidable organic carbon (%) | 0.24 ± 0.003 |
| Total N (%) | 0.03 ± 0.005 |
| Exchange cations (soluble in amonium acetate 1N pH 7) | |
| Ca (cmolc/kg) | 1.17 ± 0.092 |
| Mg (cmolc/kg) | 0.37 ± 0.015 |
| K (cmolc/kg) | 0.52 ± 0.029 |
| Na (cmolc/kg) | 0.18 ± 0.007 |
| Trace elements available (soluble in DTPA-TEA-CaCl$_2$) | |
| Fe (mg/kg) | 49.92 ± 1.245 |
| Mn (mg/kg) | 11.33 ± 0.530 |
| Zn (mg/kg) | 4.23 ± 0.118 |
| Cu (mg/kg) | 2.73 ± 0.026 |
| Texture | |
| Slime (%) | 7.18 ± 0.083 |
| Clay (%) | 3.59 ± 0.099 |
| Sand (%) | 89.24 ± 0.062 |

† Data represent the mean ± standard error ($n = 3$).

The oxidable carbon content was determined for the amendments characterization [28]. The values obtained were 46.51% for rice bran and 37.18% for the residual strawberry extrudate (Table 2). The doses were adjusted to be equivalent to each other, taking the oxidable carbon content of rice bran as a reference.

**Table 2.** Physicochemical amendment properties.

| | Oxidable c (%) | Total C (%) | Total N (%) | Total S (%) | Humidity (%) |
|---|---|---|---|---|---|
| Strawberry residual extrudate | 37.18 | 52.37 | 1.95 | 0.18 | 5.42 |
| Rice bran | 46.51 | 50.19 | 2.21 | 0.24 | 4.51 |

*2.2. Anaerobic Soil Disinfestation Trials to Evaluate the Effect of the Type of Amendment, the Dose of Amendment, and the Duration of Treatment on the Reduction of M. phaseolina Propagules*

2.2.1. Experimental Design

The experimental design was randomized blocks with three repetitions. Two trials were carried out. The only difference between them was the density of the *M. phaseolina* inoculum applied to the soil. Each trial had ten different treatments. Each treatment corresponded to an amendment type, an amendment dose, and a specific ASD duration (Table 3).

The high dose of rice bran was 20 t/ha of soil, the low dose of rice bran 13.5 t/ha of soil, the high dose of residual strawberry extrudate 25.02 t/ha of soil, and the low dose of residual strawberry extrudate 16.89 t/ha of soil. As indicated above, these doses were obtained by equalizing the amount of oxidable carbon between them. As can be seen in the list of treatments, two durations of disinfection were evaluated: 25 and 40 days.

The experimental unit consisted of a 2.5-L container filled with 4.100 kg of soil and the corresponding amendment in each case. All containers were inoculated with *M. phaseolina* at the fixed inoculum density according to the trial (see below).

**Table 3.** Treatments in the anaerobic soil disinfestation (ASD) trials.

| Abbreviation | Amendment | Dose [1] | Duration [2] |
|---|---|---|---|
| BLS | Rice bran | Low | Short |
| BLL | Rice bran | Low | Long |
| BHS | Rice bran | High | Short |
| BHL | Rice bran | High | Long |
| ELS | Residual strawberry extrudate | Low | Short |
| ELL | Residual strawberry extrudate | Low | Long |
| EHS | Residual strawberry extrudate | High | Short |
| EHL | Residual strawberry extrudate | High | Long |
| Control [3] | - | - | - |
| Metam Sodium [4] | - | - | - |

[1] Dose: Rice Bran: Low = 13.5 t/ha, High = 20 t/ha; residual strawberry extrudate: Low = 16.89 t/ha, High = 25.02 t/ha. These doses were obtained by equalizing the amount of oxidable carbon between the amendments. [2] Duration: Short = 25 days; Long = 40 days. [3] Control: without ASD treatment. [4] Metam Sodium 0.11 mL 50% w/v/kg of soil.

Both trials were carried out in growth chambers under controlled conditions that simulated the soil temperature in Huelva during the months of July and August. The chamber had a temperature of 33 °C/23 °C, a 14-h/10-h thermoperiod, and it was kept in darkness.

2.2.2. Inoculum Preparation, Soil Inoculation, and ASD Development

Rice was autoclaved in polypropylene bags with filters (PPD75/REH/V37-53, SacO2, Deinze, Belgium) according to the procedure described by Benson and Parker in 2016 [33]. Small squares with mycelium were added to sterile rice bags. The inoculated rice bags were closed and incubated at 30 °C for a week. Once the fungus had completely colonized the rice bags, the contents were grounded with a universal M20 grinder (IKA-Werke, Staufen, Germany) and sieved with a 0.425-mm sieve. The inoculum was mixed with a portion of soil previously sieved and sterilized with dry heat (using an oven at 121 °C for 8 h). The proportion soil/inoculum was one bag of rice inoculum per kg of soil. We called this mixture soil–inoculum. To ensure a homogeneous mixture of the soil–inoculum, a rotary shaker (Heidolph Reax 20, Heidolph, Schwabach, Germany) was used. To determine the inoculum density, a wet sieving [34,35] and subsequent seeding in semi-selective RB media (39 g PDA, 0.1 g rifampicin, 1 g sodium carbonate, 1 mL Ridomil Gold (Syngenta, Basel, Switzerland), 0.5 mL of Tergitol (Sigma-Aldrich, Steinheim, Germany)/1 L of distilled water) were carried out to estimate *M. phaseolina* population density counting the number of CFU (colony-forming units)/g of soil–inoculum.

The target inoculum densities selected to inoculate the soil for the trials were 700 and 2100 CFU/g of soil in trials 1 and 2, respectively. These inoculum densities were selected to ensure the development of the disease in the subsequent strawberry crop, as in strawberry fields infested with *M. phaseolina*, from 10 to more than 1000 CFU/g of soil, can be found [12,34]. A week after the inoculation, samples were taken to determine the density of the inoculum by wet sieving, as stated above. The established inoculum concentration was found to be 570.8 and 910.5 CFU/g of soil in trials 1 and 2, respectively.

Amendments and the water necessary to reach the field capacity were mixed with the soil according to each treatment [16]. The resulting mixture was placed in a black polyethylene bag that was sealed and placed in a container.

Periodic measurements of the redox potential were made throughout the process using the SensoLab benchtop pH/ORP meter (PM1000) with the ORP1000 polycarbonate laboratory ORP sensor (Sensorex Corporation, Garden Grove, CA, USA). The ORP reading in mV was converted to Eh mV by adding 201 mV [36]. To compare the intensity of the anaerobic conditions, the cumulative Eh mV hours under 263.6 mV were calculated for each pot using hourly averages of soil Eh. The value of 263.6 mV was selected as the threshold below which soil is considered as anaerobic at a soil pH of 5.55 [37]. This threshold was calculated based on the critical redox potential (CEh = 595–60 mV (soil pH)) [38].

After 25 and 40 days of ASD, samples were taken from all containers to measure the soil pH [24], and the soils were aerated for two weeks.

### 2.2.3. *Macrophomina phaseolina* and *Trichoderma* spp. Population Density Determination

After aeration, the samples were collected to assess the density of *M. phaseolina* and *Trichoderma* spp. propagules remaining in the soil after ASD following the same protocol for wet sieving as indicated in the previous section.

Plates were incubated for one week at 30 °C. Colonies morphologically corresponding to *M. phaseolina* and *Trichoderma* spp. were quantified.

### 2.2.4. Enzyme Activity Determination

β-glucosidase activity was measured according to Bandick and Dick [39]. After aeration, 350 g of soil samples were taken from each container and incubated for 15 days in opaque bags with ventilation after the addition of 37 mL of water to reactivate the activity. Subsequently, 1 g of soil was weighed, and the protocol was followed with 3 replicates per sample.

### 2.3. *Trials to Evaluate the Effect of Amendment Type, Amendment Dose, and ASD Duration of Treatments on the Severity of Charcoal Rot Disease*

#### 2.3.1. Experimental Design

Each treatment repetition in both ASD trials was transferred to 3 × 0.65 L pots. A strawberry plant ('Rábida FNM' cultivar) was transplanted into each pot (3 pots × 3 repetitions = 9 pots per treatment). These pots remained in the growth chamber under controlled conditions for 60 days to assess the severity of the disease. Pots were at 30 °C during the day and 22 °C during the night with a photoperiod of 14/10 h.

The disease severity was evaluated as a proportion of symptomatic leaves. A lack of turgor and wilting were considered leaf symptoms. The scale ranges from 0 (healthy plant) to 1 (dead plant or all leaves symptomatic), depending on the portion of tissue affected (f.e.; 0.3 = 30% symptomatic leaves). Weekly measurements of disease severity were taken from the onset of symptoms to determine the progression curve of the disease. The standardized area under the disease progress curve (AUDPCs) was calculated to integrate the measurements made during the trials [40]. The last measurement was done 50 days after the start of the trials.

At the end of the trial, severity measurements were taken as a proportion of diseased tissue from the crowns, roots, and crown vessels of the plants. To assess the crown tissue, each crown was cut longitudinally, and the severity of the crown symptoms was assessed based on a proportional scale depending on the portion of affected tissue, where 0 = no discolored tissue, and 100 = all discolored. Discolored tissue ratings were made relative to the control plants. The vascular tissue and the roots were assessed the same way as the crown, and the same scale was used. Crown and root pieces were incubated in Petri dishes with PDA (Potato Dextrose Agar) medium to confirm the causal agent.

#### 2.3.2. Data Analysis

ANOVA and mean comparison tests were conducted to determine the ASD treatment effect on: (i) the population density of *M. phaseolina* and *Trichoderma* spp., beta-glucosidase activity, cumulative anaerobiosis and pH in soil and (ii) the foliar, crowns, vessels, and root disease severity. Data from repetitions of the two trials were combined after checking that a nonsignificant trial factor (both inoculum doses of *M. phaseolina* applied behaved the same) or interaction between trial × ASD treatment occurred. In all cases, data were tested for the normality and homogeneity of variances, and when necessary, the data were transformed. Treatment means were compared according to the LSD whole significant difference test ($p < 0.05$). Mortality was analyzed by the chi-square test for multiple comparisons of proportions ($p < 0.05$) [41], which considered the observed and expected

frequencies of dead plants. The software used was Statgraphics Centurion 18 (Manugistics Inc.: Rockville, MD, USA).

## 3. Results

### 3.1. Effect of ASD Treatments on the Populations of Macrophomina phaseolina and Trichoderma spp.

All rice bran ASD treatments reduced the population density of *M. phaseolina* in comparison to the control, and the low dose with long duration rice bran treatment registered a lower density than the chemical treatment. In relation to the residual strawberry extrudate, all treatments reduced the propagules of the pathogen with respect to the control without presenting significant differences between them. Only high-dose residual strawberry extrudate treatments achieved a greater reduction in *M. phaseolina* propagules than the chemical treatment (Figure 1).

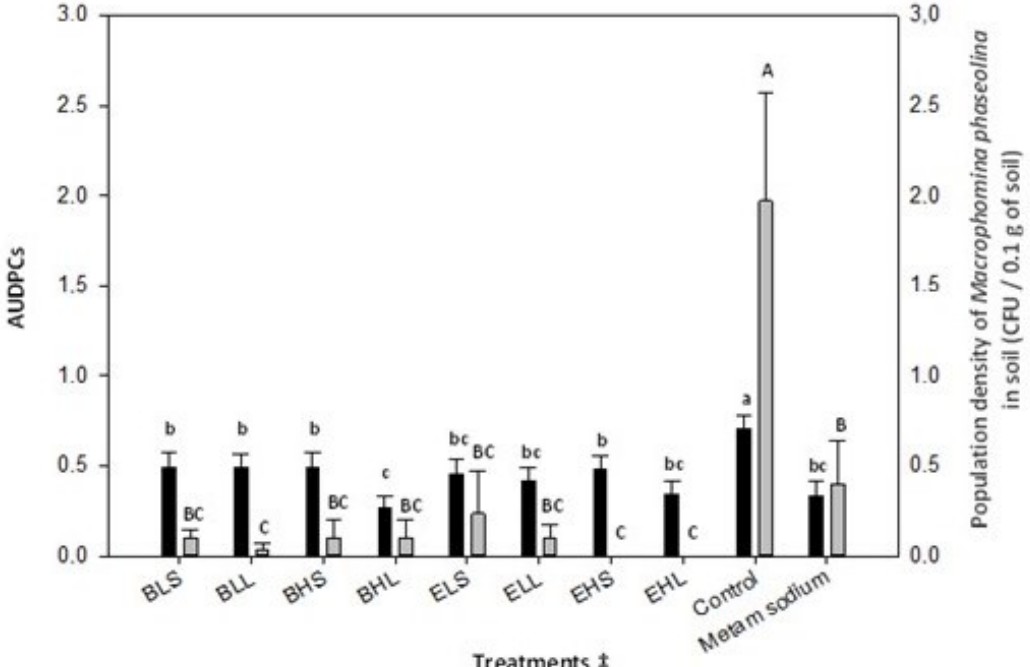

**Figure 1.** Effect of treatment on plant disease foliar severity as the standardized area under disease progress curve (AUDPCs) and population density of *Macrophomina phaseolina*. Black bars represent the strawberry plant disease foliar severity as the standardized area under disease progress curve (AUDPCs), and grey bars represent the population density of *M. phaseolina* in soil after ASD treatments. The lines above the bars represent the standard errors. Different lowercase or uppercase letters in the columns indicate significant differences according to the ANOVA test, followed by the LSD test ($p < 0.05$). AUDPC data were not transformed for ANOVA, and the population density of *M. phaseolina* in soil data was transformed with $(x + 1)^{0.5}$; $n = 18$ and $n = 6$ for both variables, respectively. ‡ Treatments are shown in Table 3.

The treatment with high dose and long duration is the one that registered the highest level of CFU/g of soil of *Trichoderma* spp., among the treatments amended with residual strawberry extrudate. The rest of the treatments with residual strawberry extrudate did not present significant differences between them. Considering the treatments amended with rice bran, most of them had lower populations than the treatments with residual strawberry extrudate. All treatments showed a higher density of *Trichoderma* spp. propagules than the control, except for rice bran treatments with short durations and the chemical treatment (Table 4).

**Table 4.** Effect of treatments on the physicochemical and biological characteristics of the soil.

| Treatment † | CA [1] (mV h) | pH [2] | Beta-glucosidase Activity (µg of p-nitrophenol/g·h) [3] | Population Density of *Trichoderma* spp. (CFU/g of Soil) |
|---|---|---|---|---|
| BLS | 5593 ± 247 c | 8.045 ± 0.051 a | 9.070 ± 1.49 bc | 119.5 ± 53 cde |
| BLL | 97,504 ± 25,290 ab | 7.805 ± 0.193 bc | 6.237 ± 1.25 bcd | 177.5 ± 60 bc |
| BHS | 44,891 ± 18,555 bc | 7.812 ± 0.094 ab | 14.080 ± 2.73 a | 15.4 ± 5 de |
| BHL | 87,747 ± 13,634 ab | 7.767 ± 0.155 bc | 5.567 ± 0.93 cd | 244.2 ± 89 c |
| ELS | 58,934 ± 17,604 b | 7.367 ± 0.131 d | 7.990 ± 0.88 bc | 531 ± 89 b |
| ELL | 77,058 ± 21,476 ab | 6.878 ± 0.101 e | 10.634 ± 1.88 ab | 510.1 ± 93 b |
| EHS | 48,458 ± 10,229 bc | 7.461 ± 0.106 cd | 10.276 ± 1.78 ab | 489.2 ± 41 b |
| EHL | 118,703 ± 43,171 a | 6.706 ± 0.075 e | 7.402 ± 1.50 bcd | 769.2 ± 103 a |
| Control | ‡ | 6.448 ± 0.040 f | 3.495 ± 1.20 d | 22.7 ± 6 de |
| Metam Sodium | ‡ | 6.738 ± 0.083 e | 5.070 ± 1.01 cd | 4.7 ± 2 e |

[1] Cumulative anaerobiosis. [2] pH measurements were performed just after the end of the ASD. [3] Beta-glucosidase activity (µg of p-nitrophenol/g·h): Measurements were taken after soil aeration. The columns represent, from left to right, the treatments, the mean values ± standard error of cumulative anaerobiosis (CA), pH, beta-glucosidase activity, and population density of *Trichoderma* spp. in soil after ASD treatments, respectively. Different letters in the columns indicate significant differences according to ANOVA, followed by the LSD test ($p < 0.05$), $n = 6$ for the variables CA, beta-glucosidase activity, and population density of *Trichoderma* spp. and $n = 12$ for the variable pH. Data were transformed with $x^{0.5}$ for CA and $x^{-3}$ for pH. † Treatments are shown in Table 3. ‡ Values not determined.

All the amended treatments reduced the density of the propagules of *M. phaseolina* against the control and increased the population of *Trichoderma* spp. naturally present in the soil in most cases.

*3.2. Effect of ASD Effect on the Physicochemical Characteristics of the Soil*

Longer ASD duration increased the anaerobic conditions in the low-dose rice bran amended treatment and high-dose residual strawberry extrudate treatment. An amendment dose did not increase the anaerobic conditions in any case. In relation to the treatments with same dose of amendment and same duration of ASD but different amendment type, just the low-dose strawberry treatment with a short duration registered more anaerobiosis compared to the low-dose rice bran treatment with short duration (Table 4).

Short duration treatments showed higher pH than the long duration ones, except for the high-dose rice bran treatments. A dose amendment did not affect the pH. For the same duration and amendment dose, rice bran showed higher pH than the residual strawberry extrudate (Table 4).

Referring to beta-glucosidase activity, all amended treatments with a short duration improved enzymatic activity compared to the control. For each dose and amendment type, the ASD duration did not show an effect on this microbial activity, except for high-dose rice bran treatments. These treatments increased the beta-glucosidase activity with a shorter duration. The effect of the amendment dose was remarkable between the rice bran and short duration treatments. Additionally, there was not any effect of the amendment type for the treatments with the same dose and duration (Table 4).

*3.3. Effect of ASD Treatments on Plant Disease Foliar Severity (AUDPCs)*

Between rice bran ASD treatments, the high-dose and long-duration treatment was the one that showed the lowest severity, although all reduced the disease severity recorded in the control and registered severities similar to the chemical treatment (Figure 1).

In treatments with residual strawberry extrudate, no effect of dose or time was observed, since the treatments did not present significant differences between them. Even so, all treatments reduced the foliar severity relative to the control and registered severities similar to the chemical treatment (Figure 1).

*3.4. Effect of ASD on Crowns, Vessels, Root Severity, and Plant Mortality*

Only high doses of both amendments reduced the crown disease severity in relation to the control. Low doses of the amendments and the chemical treatment presented similar disease severities as the control (Table 5).

**Table 5.** Effects of treatments on the crown, root, vessel severity, and plant mortality.

| Treatment † | % Crown Disease Severity | % Root Disease Severity | % Vessels Disease Severity | % Plant Mortality ‡ |
|---|---|---|---|---|
| BLS | 79.4 ± 8.4 abc | 91.4 ± 4.9 abc | 83.1 ± 7.3 a | 44, 44 b |
| BLL | 81.7 ± 8.4 ab | 93.1 ± 4.6 ab | 81.9 ± 8.5 ab | 38, 89 bc |
| BHS | 76.4 ± 8.3 bc | 91.7 ± 3.4 abc | 77.2 ± 8.0 ab | 44, 44 b |
| BHL | 62.2 ± 9.5 c | 71.1 ± 7.5 d | 62.8 ± 9.3 b | 11, 12 e |
| ELS | 89.7 ± 7.1 ab | 94.2 ± 3.5 ab | 93.9 ± 4.2 a | 44, 44 b |
| ELL | 93.3 ± 5.3 ab | 88.9 ± 5.9 abc | 93.8 ± 5.3 a | 27, 78 cd |
| EHS | 93.3 ± 4.6 b | 93.6 ± 4.1 ab | 90.6 ± 6.3 a | 33, 33 c |
| EHL | 77.5 ± 7.8 bc | 76.4 ± 7.3 cd | 76.4 ± 7.8 ab | 22, 22 d |
| Control | 94.7 ± 5.3 a | 98.3 ± 1.7 a | 95.6 ± 4.4 a | 77, 78 a |
| Metam Sodium | 81.4 ± 7.9 abc | 83.6 ± 5.2 bcd | 82.1 ± 7.3 ab | 22, 22 d |

The columns represent the treatments and the mean severity values together with their standard errors recorded for the crown, roots, and vessels expressed as percentage. Different letters in the columns indicate significant differences according to ANOVA, followed by the LSD test ($p < 0.05$), $n = 18$ for the variables % crown disease severity, % root disease severity, and % vessels disease severity. Data were transformed with $x^{2.5}$ for % crown disease severity and $x^{4.5}$ for % root disease severity. † Treatments are shown in Table 3. ‡ Percentage of plants dead by *M. phaseolina* ($n = 18$) 50 days after planting in soil where ASD treatments were applied. In the column, mean values followed by the same letter are not significantly different according to the multiple comparisons for proportions test, $p < 0.05$.

Just the ASD treatments with high doses and long durations, along with the chemical one, reduced the root disease severity compared to the control. The duration of ASD had a remarkable effect on high-dose amended treatments, reducing root rot on strawberry plants.

The treatment with high-dose rice bran and a long duration was the only one that registered less disease severity in the vessels than the control. The rest of the treatments, including the chemical treatment, did not present significant differences from the control (Table 5).

The rice bran treatment at a high dose and long duration reduced the severity with respect to the control in the crown, roots, and vessels. Nevertheless, the treatment with residual strawberry extrudate at high doses and a long duration only registered less severity in the crown and roots (Table 5).

Referring to the severities and survival of *M. phaseolina*, we observed that, at equal doses and ASD duration, there were no differences between the two amendments. The dose effect was only observed in the case of rice bran with a long ASD duration, reducing the leaf, crown, and root severity. Additionally, a time effect was only observed for high-dose rice bran treatments in foliar and root severity and for high-dose residual strawberry extrudate treatments in root severity.

The control treatment showed the highest percentage of mortality among all treatments. In the treatment with rice bran, a high dose and long duration showed the lowest mortality registered, achieving a lower percentage of mortality than the Metam Sodium treatment. After treatment with rice bran at a high dose and long duration, the strawberry residual extrudate and long duration treatments (for high and low doses) also show low mortality rates equal to Metam Sodium.

## 4. Discussion

Previous studies have proven the efficacy of ASD for disease control in different crops, such as apple, tomato, okra, eggplant, and strawberry [42–44]. Specifically in strawberry crops, the efficacy of anaerobic soil disinfection with rice bran as an amendment has already been proven to reduce plant death due to *M. phaseolina* [43].

Other studies evaluate the efficacy of ASD in reducing soil pathogen propagules or examine the reduction of disease in strawberry plants after the soil has experienced an ASD treatment [43,45,46]. In contrast to that, a noteworthy aspect of this study is the continuous monitoring of the effects of ASD on the survival of the propagules and whether this reduction of propagules, together with the modification inflicted on the soil microbiota, translates into a reduction of the disease. In short, the soil was inoculated, treated with ASD, and used for growing strawberries in a growth chamber, thus allowing the complete process of what would happen in a real field situation to be studied.

Initially, the two trials conducted were inoculated with different pathogen doses, but after the statistical analysis, there was no difference in disease pressure between the trials. This was due to the reduction of propagules during the establishment time (data shown in the Inoculum Preparation, Soil Inoculation, and ASD Development section). This loss in viability of part of the inoculum reduced the difference between the initial inoculum doses of the trials so that both eventually reached an equivalent disease pressure.

After ASD treatment, variations in the *M. phaseolina* and *Trichoderma* spp. populations were observed. The observed reduction of *M. phaseolina* population density may be caused, among other factors, by the presence of *Trichoderma* spp. in the soil, since its population density increases in most of the treatments. This could indicate the potential of *Trichoderma* spp. to act as a biological control against *M. phaseolina*, as has also been highlighted by other authors in the past [47–50]. In fact, there is evidence of the efficacy of some treatments that incorporate *Trichoderma* spp. as a biological control against *M. phaseolina* in strawberries. Some of these treatments can reduce charcoal rot caused by this pathogen by up to 44% under controlled conditions and 65% under field conditions [51], which is consistent with the results obtained in this study.

The treatment with the highest density of *Trichoderma* spp. propagules corresponds to high-dose residual strawberry extrudate with a long duration. This highlights the importance of the dose of the amendment and time in the case of residual strawberry extrudate to promote the growth of *Trichoderma* spp. There is evidence that the use of amendments promotes the appearance of microorganisms of biological control [49]. These, in turn, are able to reduce the number of pathogenic fungal propagules present in the soil and achieve a drastic reduction of the disease in strawberry plants [49].

With the trials carried out in this study, all treatments have been proven to be adequate for reducing charcoal rot caused by *M. phaseolina*. Part of these results are consistent with those obtained in other studies where rice bran is also used at 20 t/ha (high dose) to perform ASD in order to control *M. phaseolina* in strawberries [31,43]. Muramoto et al. 2016 [43] achieved a 38.64% plant mortality reduction with rice bran (20 t/ha) and 27 days of ASD treatment compared to the growing standard in a *M. phaseolina*-infested organic field. Instead, we achieved an 85.71% plant mortality reduction with the rice bran ASD treatment at the same dose and 40 days. The main difference, besides the experimental scale, between both trials was the treatment duration. This underlines the importance of the duration of ASD treatments and their impacts on soil suppressiveness. In rice bran treatments, longer times seemed to improve the suppressiveness against the disease compared to the shorter duration by unknown mechanisms.

The duration of the treatment does not seem to influence the disease reduction in the case of residual strawberry extrudate, since all treatments containing this amendment perform similarly to the fumigant and better than the control. This could indicate that residual strawberry extrudate is a good alternative to chemical treatments for soil disinfestation. Current studies show that amendments not commonly used in ASD, such as composted chicken manufacture, can give very good results in terms of reduced severity in diseases caused by soil pathogens and can improve the productivity of strawberry plants, notably increasing the benefits perceived by farmers [52]. Residual strawberry extrudate is also an amendment whose results are promising. As an advantage, this residue can be found near the strawberry growing area, and its reintroduction at the beginning of the process increases the value of the waste. The use of residual strawberry extrudate in this crop

would be a clear example of food waste circular economy. Residual strawberry extrudate is generated in the manufacturing process of products such as yogurts, jams, and other byproducts. Approximately 21% of the world's strawberry production is used in these processes. At the end, 7% of the initial manufactured strawberry becomes a residue. In summary, 1.47% of the world's strawberry production ends up as residual strawberry extrudate. By using this residue for ASD, we would be promoting its use in the sector and minimizing discharges of high organic load into the environment [32].

The reduction of propagules in the soil, together with the suppressive environment caused by ASD treatments, corresponded to a subsequent reduction of the disease on strawberry plants. The generation of this suppressive environment was influenced by various factors such as pH, REDOX potential, temperature, and microbial activity, among others.

Soil pH is an important factor to consider as a condition in the development of ASD and the enzymatic reactions that take place during the disinfestation. The pH can vary according to the chosen amendment and the sampling time (from pretreatment to post-treatment) [53]. The acidification of the soil pH after ASD treatment could be due to the release of volatile organic acids during the disinfestation process. The presence of both acetic and butyric acids during the anaerobic decomposition of wheat bran during ASD treatment and the resulting reductions in soil pH was reported by Momma et al. 2006 [21]. Lower soil pH values were related to higher values of cumulative anaerobiosis (CA), as the longer the anaerobic conditions are prolonged, the more fatty acids are released into the process [53]. In summary, low soil pH after ASD treatment is a potential indicator that organic acids, which are important to disease control, are being created through the anaerobic decomposition of the amendments [54]. Our results indicated that the treatments with residual strawberry extrudate and long durations were the ones with the lowest pH among the amended treatments, and this would translate into a higher production of volatile fatty acids. The rice bran treatments showed high pH, especially the short duration treatments. This could indicate a lower release of fatty acids or a higher release of cations, including ammonium [43], during organic matter decomposition [53], which could be masking the acidification of the soil. In this sense, the amendment dose seemed to have no influence on the resulting pH, but the type of amendment and the ASD duration did.

Soil temperature is a conditioning factor for the effectiveness of ASD [16,55], which is reduced in soils where low temperatures are reached [46]. In these trials, the temperature was set at 33–23 °C during ASD, because this is the estimated average temperature in the soil of the strawberry growing area of Huelva between July and August (period in which ASD was performed). Based on the results obtained, this temperature was shown to be suitable for the ASD against charcoal rot in strawberries.

REDOX potential and, specifically, the CA are also important to the ASD development. This parameter was measured in ASD trials to check if the necessary reducing environment was achieved. A previous study stablished that at least 50,000 mV h are needed for ASD to be effective against *V. dhaliae* [37]. In the case of *M. phaseolina*, the exact threshold for the control is not yet known [43]. In both trials, there were treatments that did not reach this threshold or only slightly exceeded it. These treatments corresponded to those of a short duration, regardless of the amendment. The treatments with long durations exceeded the 50,000-mV h threshold, especially those with a high dose of residual strawberry extrudate. The CA achieved by the amended treatments resembled those achieved in publications whose trials were of similar durations to the present study [37]. Thus, the ASD duration can be the key in the achievement of the CA required by other authors for soil disinfestation.

In these trials, all the treatments controlled the disease caused by *M. phaseolina* and reduced the propagules even without reaching the threshold determined by Shennan et al. 2018 for *V. dahliae* [37]. This states that the threshold needed to control charcoal rot caused by *M. phaseolina* was lower, which would allow for the use of a shorter duration and lower amendment doses in the ASD treatments to achieve the same purpose. The REDOX potential, with a precisely established threshold, could be a good indicator of the effectiveness

of ASD. Both residual strawberry extrudate and rice bran had an adequate capacity to promote the anaerobiosis conditions necessary in ASD and the subsequent propagules and disease severity reduction.

The resulting soils showed variations on the microbial activity. This activity could influence the suppressive environment. Beta-glucosidase is a C-acquiring enzyme [53], and its presence is indicative of the activity generated by the microbial community of the soil. This enzyme is very sensitive to changes in pH and soil management practices [56]. The increase of beta-glucosidase activity in rice bran amended treatments with a short duration and high dose could be caused by the remaining carbon source still available in the soil. With a shorter duration, the carbon source was not yet consumed, and therefore, the microbiota was more active. Generally, in amended treatments with a longer duration, the microbial activity recorded was lower, probably because the amount of carbon reduced over time as it was consumed. Additionally, enzyme activities in amended soils tended to decrease over time [57]. Some studies affirmed that the performance of ASD promoted the increase of beta-glucosidase activity [29]. The importance of this activity in the reduction of pathogen propagules and its dependence on soil pH have been demonstrated before [29]. There have also been studies on the ability of *Trichoderma* spp. to produce beta-glucosidase after feeding on plant residues [58]. The activity produced by this biological control fungus is capable of inhibiting *M. phaseolina* growth in vitro [58].

In summary, residual strawberry extrudate and rice bran were established as good candidates to obtain adequate disinfestation results.

Residual strawberry extrudate is a waste product with low cost. In Coastal California, the main constraint for ASD treatments in strawberry farms is the rising cost of rice bran as a carbon source [59]. In addition, residual strawberry extrudate generation coincides with the end of the strawberry season, making the timing ideal for its use. The ASD treatment is applied after the plants' removal at the end of the season, so the generation of the residual strawberry extrudate coincides with the field application. This makes the residual strawberry extrudate an even better option than rice bran for use in ASD treatments on strawberry cultivation. Furthermore, the residual strawberry extrudate used as an amendment in ASD at the same dose (oxidable carbon) and same duration has the same effectiveness in reducing propagules and disease severities as rice bran. In the case of rice bran, the ASD effectiveness could be improved if farmers had the possibility to use higher doses and longer durations of treatment, although this would require a prior economic study. In this way, further research needs to be done to understand the underlying mechanisms and optimize the available resources, such as amendments, water, and time, in order to minimize the costs and create affordable commercial treatment strategies for the sector.

Anaerobic soil disinfection is proving to be a very promising method of disease control in a wide variety of crops worldwide [42–45]. Even so, new trials in this direction need to be validated to check whether these results obtained in the highly controlled growth chamber can also be achieved under field conditions. The environmental and spatial variability is much wider under these conditions. In the field, the homogeneous distribution of the C-source and water both spatially and in depth on the field scale greatly impacts the subsequent disease development. For this purpose, new investigations will be carried out directly in the fields to be disinfested.

## 5. Conclusions

- All treatments amended with residual strawberry extrudate and rice bran resulted in a marked reduction in *M. phaseolina* propagules in the soil.
- Both residual strawberry extrudate and rice bran cause a reduction of charcoal rot caused by *M. phaseolina* when used as amendments in ASD.
- The resulting soil disinfestation against *M. phaseolina* with the residual strawberry extrudate at 16.89 t/ha and 25 days was similar to the most effective treatment with rice bran (20 t/ha and 40 days) based on the disease foliar severity.

    — Behind the suppressive effect of ASD against *M. phaseolina*, the action of *Trichoderma* spp. as a biological control agent could be operated. The growth of this microorganism is favored by treatments with residual strawberry extrudate.

**Author Contributions:** Conceptualization, M.A. and C.B.; methodology, M.A., C.B. and A.M.-C.; formal analysis, M.A.; investigation, M.A., C.B., A.M.-C and P.H.-M.; data curation, A.M.-C.; writing—original draft preparation, A.M.-C.; writing—review and editing, M.A. and C.B.; supervision, M.A. and C.B.; project administration, M.A.; and funding acquisition, M.A. and C.B. All authors have read and agreed to the published version of the manuscript.

**Funding:** Grant RTI2018-094537-B-I00 funded by MCIN/AEI/10.13039/a501100011033 and by ERDF A way of making Europe.

**Institutional Review Board Statement:** Not applicable.

**Informed Consent Statement:** Not applicable.

**Data Availability Statement:** The data presented in this study are available on request from the corresponding author.

**Acknowledgments:** We thank Silvia Pérez and Javier Ordóñez for the excellent technical assistance.

**Conflicts of Interest:** The authors declare no conflict of interest. The funders had no role in the design of the study; in the collection, analyses, or interpretation of the data; in the writing of the manuscript; or in the decision to publish the results.

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
