# Peer review of "Use Optimization of Organic Wastes in Anaerobic Soil Disinfestation against Strawberry Charcoal Rot Root"

_horticulturae, doi:10.3390/horticulturae8090841_

Round 1
Reviewer 1 Report
This is a well-written paper on a carefully and thoughtfully designed study. Authors examined the efficacy of anaerobic soil disinfestation (ASD) with different carbon sources, rates, and durations on the control of Macrophomina phaseolina, the causal pathogen of charcoal rot, followed by a bioassay using strawberry crop in a controlled environment. Populations of M. phaseolina and Trichoderma spp. and ß-glucosidase activity in the soil were measured along with the disease severity of strawberry plants. The experiment was repeated twice, providing conclusive promising results.
The study found residual strawberry extrudate as an effective carbon source for ASD, a novel and excellent use of strawberry waste for material recycling, and cost-effective soil-borne disease control.
I look forward to the outcomes of field trials based on this study.
Below are minor comments;
Line 13: “against” should be non-italic.
Line 13: Replace “an alternative more sustainable” with “a more sustainable alternative”.
Table 1: In Texture, replace “slime” with “silt”.
Line 107: Add a reference for “oxidable carbon” method used.
Line 132: I suggest adding “(see below)” after “to the trial.”
Line 154: I suggest adding “target” between “The” and “inoculum densities”.
Line 175 and many other lines afterward: Make species names Italicized (e.g., M. phaseolina, Trichoderma, V. dahliae)
Line 317: A reference 42 instead of 39?
Line 356: Insert “Besides the experimental scale (i.e., a field trial vs. pot trials)” before “the main difference”
Lines 359-360: Why is this the case? If you can think of any mechanisms, insert after line 360. Otherwise, I would suggest inserting “by unknown mechanisms” after “shorter duration”.
Line 399: Add “including ammonium” after “cations” and add a reference 42.
Line 446: Add “bran” after “rice”.
Author Response
Dear Reviewer,
We deeply appreciate the time you have taken to review this article. We are flattered that both the manuscript and the research were to your liking. All proposed changes have been checked and incorporated into the text. With these modifications we hope to improve the quality of the manuscript.
We remain at your disposal for any other suggested changes,
Best regards.

Reviewer 2 Report
The paper provides some interesting findings related to ASD and control of Macrophomina charcoal rot in strawberries. The results are of interest and merit, but aspects of the manuscript need to be improved before it is acceptable for publication. In general, the paper is quite confusing to read, and could do with some editing for improved English, sentence structure and logical flow of the points being made. Further, insufficient information is provided in the methods for some key measurements as discussed below.
The major weakness of the paper as written is the lack of clarity on the effectiveness of the disease reductions observed with the various ASD treatments in terms of empirical plant performance measures such as biomass, berry yield, or plant mortality. Without this it is hard to justify the statement made at the end of the abstract (line 22) and in the text that ASD with strawberry extrudate “is good enough to disinfect the soil against M. phaseolina. Yes, the soil propagule density is greatly reduced, but to say it is good enough implies that the disease is well controlled. It is impossible to interpret what the differences in foliar disease measure of AUDPCs really mean as a measure of impact on plant productivity. No information is provided on the scale used in the measurement and there is no way of knowing what the difference between plants with an AUDPC of around 0.7 (control) and 0.4 (some ASD treatments) is in a productivity or economic sense. There are similar issues with the data in table 4 on disease severity in crown, root and vessel tissue. Simply stating that the numbers mean the proportion of diseased tissue is insufficient, how is this actually measured – what constitutes diseased tissue in each case. The proportion of diseased tissue at the end of the experiments is very high in all treatment, with the best control being a reduction from 95% plus to around 68-75% for the best ASD treatments. Since it is stated in the discussion that a 65% reduction in plant mortality was achieved with some rice bran ASD (exact treatment not specified) such a modest reduction in diseased tissue was related to greater plant longevity. At a minimum the plant mortality data needs to be presented, but ideally biomass and fruit yield data should also be included. How are plant growth and yield data related to measures of soil propagule density of % diseased tissue – are they correlated, or is there a threshold of soil propagule density above which disease severity becomes sufficient to reduce growth and increase mortality? Which of the measures of disease incidence are the best predictors of impacts on plant growth, productivity and survival?
Similarly, it would be interesting to look more closely at the relationships between Trichoderma numbers and disease severity – is there any correlation?
The dangers of extrapolating the performance of ASD under highly controlled growth chamber experiments to its likely effectiveness under field conditions needs to be discussed. While duration of ASD, temperature and C-source type and rate are all important variables, performance in the field depends on other factor. In particular, the distribution uniformity of C-source and water both spatially and with depth on a field scale greatly impacts subsequent disease development. For example, if ASD is applied just to beds, the untreated furrows can be a source of pathogen invasion; or if anaerobic conditions are not created everywhere due to lack of irrigation uniformity or soil spatial variability, pockets of pathogen propagules can survive and subsequently infect the crop.
It would be helpful to know more about the strawberry extrudate in terms of its physical form and its chemical analysis. The nitrogen provided by rice bran helps stimulate early season growth in strawberry which contributes to the economic performance of ASD, what is the N content of the extrudate?
An interesting study that merits publication if these changes are made
Author Response
Dear reviewer,
First of all, we deeply appreciate the time spent reviewing this manuscript. Each and every one of the revisions, questions and suggested changes have been carefully addressed and answered. We believe that after including the appropriate modifications, the manuscript has significantly improved in quality.
For future changes or suggestions we are entirely at your disposal,
best regards.
